# Investigation into Time-Dependent Deformation and Recovery Rates in Non-Irradiated and Irradiated Polymers

**DOI:** 10.3390/ma18102316

**Published:** 2025-05-16

**Authors:** Anatoliy I. Kupchishin, Marat N. Niyazov, Sergey A. Ghyngazov

**Affiliations:** 1Physico-Technological Center, Abai Kazakh National Pedagogical University, Dostyk, 13, Almaty 050010, Kazakhstan; ankupchishin@mail.ru; 2Al-Farabi Kazakh National University, Al-Farabi Ave., 71, Almaty 050040, Kazakhstan; 3Research School of High Energy Physics, Tomsk Polytechnic University, Lenin Ave., 30, Tomsk 634050, Russia; ghyngazov@tpu.ru; 4Department of Computer Science, Tomsk State Pedagogical University, Kievskaya Street, 60, Tomsk 634061, Russia

**Keywords:** polyethylene, polytetrafluoroethylene, irradiation, mechanical properties, exponential model, linear model, irradiation dose, strain rate, strain recovery rate

## Abstract

The aim of this study was to investigate the effect of dose, temperature and mechanical stress on the return and rate of return deformation. The structure of the polymers under study begins to change under the influence of electron and ion beams, as well as temperature and mechanical stress, which leads to a change in such mechanical properties as return deformation and rate of return deformation. The authors proposed formulas for models that accurately explain the experimental data. In addition, using an optical microscope DM 6000M (Leica, Wetzlar, Germany), photographs of the surface morphology of unirradiated and irradiated materials before and after tensile testing were obtained. The results can be used to improve the properties of packaging materials in the food industry.

## 1. Introduction

It is impossible to imagine modern society without the use of polymers in various branches of life. Half a century ago, objects made of wood, metal, and fabric were a priority for people at home, at work, etc. However, now polymers such as polyethylene (PE), polytetrafluoroethylene (PTFE, fluoroplastic), and polyimide (PI) have become the main substances used by people [1]. After obtaining polymeric materials, i.e., substances with a high molecular weight, their various properties began to be actively studied. The study of the deformation and strength properties of polymeric materials, as well as their modification by various methods and techniques, is no exception. The work [2] examines the physical and chemical processes, as well as the mechanical characteristics of materials implanted with nitrogen and helium ions with an energy of 100 keV, which were exposed to irradiation. In the study [3], the structural properties of both primary and irradiated polymer insulation were assessed using scanning electron microscopy (SEM). These properties were found to be closely related to the effects of ion irradiation and the structure of the material.

One of the current and effective methods for changing the mechanical properties of materials, including polymers, is the use of ion and electron beams to irradiate the object of study. Ion and electron beams have an advantage due to their ability to provide the maximum energy loss density among all types of radiation (gamma rays, electron and ion beams) [4]. Considering that irradiation is actively used to reduce friction and wear, as well as to increase the strength and plasticity of materials with minimal impact on the bulk properties of various polymers, it becomes justified to study the effect of irradiation on polyethylene and other polymer films and their structural and mechanical characteristics. Reviews concerning the state of knowledge about the properties of irradiated polymers can be found in [5]. Irradiation causes shrinkage and smoothing of the surface layer of the polymer, which suggests that these effects may be associated with the release of hydrogen from the surface layer. Radiation technologies for polymer processing include a wide range of methods using various types of radiation and sources, as well as various materials, and solve a variety of applied problems [6].

The presence of various radioactive sources causes various phenomena that lead to defects in the structure of materials and changes in their properties, including mechanical ones. Therefore, the study of the effect of the absorbed radiation dose on the characteristics of materials remains a topical issue for many researchers [7]. One of the popular polymers is polyethylene (PE), which is widely used in the nuclear and aerospace industries, as well as for the production of cables, missile shields, fuel pumps and other products [8]. Studies of the effect of low-dose irradiation (up to 200 kGy) on various polymers have shown that changes in the degree of crystallinity and tensile strength depend on the irradiation dose and are due to the competition between the processes of crosslinking and chain division in the polymer structure [9]. To modify the surface of polymers, the effects of PDA coating and subsequent introduction of functional groups are considered in [10,11]. Many studies have been conducted using experimental and computational methods to reduce wear and study the mechanisms of deformation. The processes of deformation and mechanical destruction are considered in [11].

In the literature [12,13,14,15,16], the authors study the rate of return deformation of such materials as polyimide, glass and various alloys irradiated with protons. The authors come to the conclusion that the improvement of deformation characteristics by various methods, such as irradiation and others, will help to improve the properties of structural parts of various devices and constructions.

There are many works devoted to the study of the mechanical properties of non-irradiated and irradiated polymeric materials, but very few papers devoted to the study of the effect of the radiation dose on the rate of deformation and the rate of return deformation (Vε) of polymers under constant load, which is the purpose of this study.

## 2. Experimental Procedure

The characteristics of the materials for this study, the type and modes of irradiation are presented in Table 1.

The film sheets were cut into 5 mm wide strips using a special cutting device that allowed the width and length of the samples to be cut to be adjusted. The working length of the samples was 5, 7, 10 and 12 cm. Some of them were not irradiated and were used as controls. The samples to be irradiated were located at a distance of 300 mm from the accelerator output window. Electron irradiation was performed on an ELU-6 electron accelerator with a particle energy of 2 MeV. The vacuum in the accelerator system was maintained at a level of 10^−6^ mm Hg. These modes were selected based on dose accumulation over a certain period of time. A DRG-01t1 dosimetry system was used to assess the dose distribution in the accelerator hall. The temperature conditions of this study were maintained at a level of 23 °C.

The experiments measured the rate of deformation and return deformation of the samples. The rate of deformation was calculated using the graphs of the dependence of deformation on time under static stress. And the rate of return deformation was calculated after the load on the samples was removed and they began to take their previous shape.

Tensile tests were performed according to ASTM-D882 [17] using a universal testing machine model RU-50. The tests were conducted at a stroke rate of 16 mm/min and an AC frequency of 10 Hz, which was controlled by a CHNT inverter. To investigate the relationship between elongation and stress (strain, σ) and other dependencies, we upgraded our equipment, and now it allows us to measure parameters using motion and force sensors under various loads and observe their changes over time. The setup uses an interface with motion and force sensors from Science Cube. The deformation data collection frequency is 2.5 mm⋅s^−1^.

Mechanical tests were carried out on a specially developed installation based on a tensile testing machine RU-50 with a maximum force of 50 kN. The speed of the crosshead and sample extension, respectively, was 1.6 mm/s. The rate of deformation and return deformation of the samples was measured in the experiments. The rate of deformation was calculated using the graphs of the dependence of deformation on time under static stress. And the Vε was calculated after the load on the samples was removed and they began to take their previous shape.

## 3. Results

The rate of deformation of the polyethylene under study with respect to the change in irradiation dose is a constant value for each temperature point, as can be seen from the graphs in Figure 1.

Irradiation of polyethylene with electrons (or an electron beam) causes several specific changes in its molecular structure and physicochemical properties. Electron irradiation is a type of ionizing radiation that affects polyethylene similarly to gamma and X-ray radiation, but with certain features associated with high energy and a short electron path length. The electron beams affected the polymer in such a way that cross-linking of molecules and degradation of molecules caused by molecular breakdown and oxidation did not affect the characteristic shown in Figure 1.

Figure 2 shows the dependence of the Vε as a function of irradiation dose on the absorbed dose of irradiation with electrons with an energy of 2 MeV for fluoroplastic films. The polymer sample with a thickness of 100 μm has a Vε as a function of irradiation dose greater by 3.3 times than the polymer with a thickness of 40 μm. However, with an increase in the irradiation dose, the studied parameter begins to decrease for both thin and thick samples of the material. Electron irradiation creates free radicals that can break chemical bonds in the molecular chain, and also promote depolymerization. This reduces the molecular weight of the polymer and leads to a violation of its structure. As a result, the material loses the ability to restore its shape after deformation, which reduces its elasticity.

The curves are satisfactorily described by the exponential model, the formula of which was obtained within the framework of the cascade-probability method (CPM):(1)$$\varepsilon_{D}^{|}=\;\varepsilon_{0D}^{|}\cdot (\exp(\;D/D_{0})-1).$$

In (1) $\varepsilon_{0D}^{|}$ is the limiting value of the Vε as a function of irradiation dose, $D_{0}$—the dose at which the parameter ($\varepsilon_{D}^{|}$/$\varepsilon_{0D}^{|}$ − 1) decrease by a factor of e [9,10,11]. In Formula (1), the parameter $\varepsilon_{0D}^{|}$ equal to 3 and 9 %/kGy. In Formula (1), the parameter $D_{0}$ is calculated from the formula and is equal to 2.7 and 3 kGy for each case.

Figure 3 shows the dependence of the Vε on time for non-irradiated and irradiated with krypton ions polyethylene film with a working length of 5 cm at a static stress of 80 MPa. It is evident that the Vε depends significantly on both time and static stress and the ion irradiation dose. At a constant load, there is a decrease in the Vε depending on t. At a stress of 80 MPa, it reaches a value of about 3% per second due to the straightening of randomly located chains in the irradiated sample. After this, there is a decrease in the rate due to the resistance of the material to deformation due to a decrease in the cross-sectional area of the sample, which leads to an increase in rigidity. Just as in the case of irradiation of material with high-energy electrons, irradiation of material with ions leads to a significant decrease in Vε. Irradiation can cause cross-linking of polymer molecules, which makes them more rigid and less pliable. In the case of polyethylene, this can lead to the formation of a cross-linked structure, which increases rigidity but reduces the ability to undergo plastic deformation.

The curves are satisfactorily described by an exponential model, the formula for which was obtained within the framework of the cascade-probability method (CPM):(2)$$\varepsilon_{r}^{|}=\;\varepsilon_{0r}^{|}\cdot (\exp(\;t/t_{0})-1).$$

In (2), $\varepsilon_{0r}^{|}$ is the limiting value of the Vε, which is the time at which the parameter ($\varepsilon_{r}^{|}$/$\varepsilon_{0r}^{|}$ − 1) decreases by a factor of e [9,18].

In Formula (2), the parameter $\varepsilon_{0r}^{|}$ is equal to the maximum value of the return deformation, which is obtained for each case and is, respectively, equal to: 0.6, 1.23 and 1.7%·s^−1^. And the value $t_{0}$ is calculated from the formula and is equal to 12, 14 and 16 s for the three curves, respectively. The cascade-probability method (CPM), invented by us, is used in radiation physics to calculate the concentration of radiation defects, including in materials. It is based on the cascade-probability function (CPF), which has the meaning of the probability of the primary particle to reach a depth (h) with energy (E_0_) after the number of collisions (i). In this paper, the simplest CPF is used. Within the framework of this method, we have explained many experimental data:

When polymers are irradiated, chains are mainly broken and simple defects are formed that affect mechanical properties, including deformation.When materials are irradiated with electrons with an energy of 2–6 MeV, electrical resistance, internal friction and positron annihilation at low and normal temperatures occur.

Figure 4 and Figure 5 show the dependences of the Vε on time. As in Figure 2, the Vε parameter, which in Figure 4 and Figure 5 was calculated over time, and not over the dose as in Figure 2, also decreases. For example, over a time of 30 s, this parameter approached zero. Another possible reason may be that electron irradiation can also lead to the formation of cross-links between fluoroplastic molecules. This is not always beneficial for elasticity, since in the case of excessive cross-linking, the polymer loses flexibility and becomes more rigid and brittle. Due to the formation of incorrect or disordered cross-links, the structure of the material is disrupted, and its ability to restore the original shape after loading worsens, which leads to a decrease in elasticity.

The authors conclude that the material was more plastic before irradiation with krypton ions and electrons, and therefore the rate of its return deformation was greater in the unirradiated state. However, the material becomes more rigid after irradiation with both krypton ions and electrons, which indicates an improvement in its strength properties due to the cross-linking occurring in the structure of the substance.

Optical micrographs were obtained in the transmitted light mode (black and white images) and in the dark field mode (light spots on a dark background) using different objectives with magnifications of ×5000 and ×1000. All images are accompanied by a corresponding scale bar. In the case of PE samples after testing (both before and after irradiation with 2 MeV electrons), elongated pores are clearly visible.

The images in Figure 6 show dark areas corresponding to surface contamination and volume inhomogeneities in the sample. Elongated spindle-shaped areas—pores and microcracks—are observed on the samples after mechanical testing. Similar to the unirradiated sample before testing, dark spots of inhomogeneities and/or surface contamination are observed on the images of the irradiated PE film. The samples irradiated with high-energy electrons with an energy of 2 MeV after testing also demonstrate the appearance of microcracks and elongated pores, which appear as dark areas in the bright field mode, and as light stripes on a dark background in the dark field mode. Unfortunately, optical images obtained with the microscope cannot show the areas where crosslinks and other structural defects occurred, because they occur at the nanoscale, and the resolution of this microscope was insufficient as well as other types of microscopes. However, the surface deformations of the globules along the stretching axis are quite clearly visible. The stretching was performed vertically along one axis.

The PTFE samples demonstrate a denser and less transparent structure, compared to the PE film, which is evident from the photographs of the surface image in Figure 7. In the case of non-irradiated PTFE before testing, non-uniformities are observed in the sample volume, probably associated with different densities. The photographs of the non-irradiated PTFE sample after mechanical testing show that the dark areas corresponding to denser regions have become more elongated along the film tensile axis. The sample irradiated with 2 MeV electrons in a vacuum before testing, similarly to the non-irradiated sample, demonstrates a non-uniform density distribution, which is expressed in dark areas in the micrographs. The irradiated PTFE sample after mechanical testing demonstrates more pronounced dark areas directed along the tensile axis, compared to the non-irradiated sample after testing. Analysis of microcracks and pores after stretching of non-irradiated and irradiated materials indicates that the length of microcracks decreases by 8% after irradiation with charged particles. This was expected since the return-deformation characteristics of the samples weakened.

The analyzed PTFE films demonstrate a Raman signal typical for this material, Figure 8. The peaks in the region of 290 cm^−1^ and 384 cm^−1^ correspond to torsional and deformation vibrational modes of the CF_2_ group. The most intense peak at 733 cm^−1^ is caused by symmetric vibrations of compression and extension of the C–C bond, while the pronounced peak in the region of 1380 cm^−1^ corresponds to symmetric vibrations of compression and extension of the CF_2_ group. Weakly expressed peaks at 1217 and 1300 cm^−1^ are attributed to antisymmetric vibrations of CF_2_ extension.

Similar to the PE samples, the position and width of the peaks are preserved for all samples, indicating the preservation of structural units. The dependence of the intensity of some peaks on the orientation relative to the extension axis is also observed. It is important to note that, according to literature data, the intensity of the peak in the region of 1380 cm^−1^ often correlates with the deformations of the –CF_2_– group in the polymer chain (spectra 2 and 5—along the stretching axis).

The analyzed PE films demonstrate characteristic Raman bands of polyethylene (Figure 8). The peaks at 1064 and 1128 cm^−1^ correspond to compression and stretching of C–C bonds. The line at 1296 cm^−1^ is caused by the vibrational mode of CH_2_ twisting of the methylene group. The triplet at 1440 cm^−1^ is caused by the bending of CH_2_ methylene groups. The broad peak at 2723 cm^−1^ is a combination of several vibrational modes. The peaks in the range of 2800–3000 cm^−1^ are usually responsible for compression and stretching of C–H bonds. In particular, the peaks at 2852 cm^−1^ and 2888 cm^−1^ correspond to symmetric and asymmetric stretching of the CH_2_ methylene groups, respectively. The shoulder-shaped peak at 2930 cm^−1^ is also caused by compression and stretching of the C–H bond of the methylene group. Analysis of the spectra of all PE samples before and after irradiation showed that the width and position of the peaks remain unchanged, indicating the preservation of structural units. However, the ratio of peak intensities turned out to be sensitive to the direction of film stretching, which is expressed in an increase in the intensity of peaks in the region of C–C stretching and twisting of the CH_2_ group (spectra 2 and 5—along the stretching axis). The preservation of structural bands in the Raman spectroscopy spectra of polymers indicates the preservation of the basic chemical structure and symmetry of the molecules in the studied sample. That is, there are no strong chemical transformations in these polymers—there is no strong destruction, complete cross-linking or other reactions that could change the spectrum.

Similar to the PE samples, the position and width of the peaks are preserved for all samples, indicating the preservation of structural units. The dependence of the intensity of some peaks on the orientation relative to the extension axis is also observed. It is important to note that, according to literature data, the intensity of the peak in the region of 1380 cm^−1^ often correlates with the deformations of the –CF_2_– group in the polymer chain (spectra 2 and 5—along the stretching axis).

The analyzed PE films demonstrate characteristic Raman bands of polyethylene (Figure 9). The peaks at 1064 and 1128 cm^−1^ correspond to compression and stretching of C–C bonds. The line at 1296 cm^−1^ is caused by the vibrational mode of CH_2_ twisting of the methylene group. The triplet at 1440 cm^−1^ is caused by the bending of CH_2_ methylene groups. The broad peak at 2723 cm^−1^ is a combination of several vibrational modes. The peaks in the range of 2800–3000 cm^−1^ are usually responsible for compression and stretching of C–H bonds. In particular, the peaks at 2852 cm^−1^ and 2888 cm^−1^ correspond to symmetric and asymmetric stretching of the CH_2_ methylene groups, respectively. The shoulder-shaped peak at 2930 cm^−1^ is also caused by compression and stretching of the C–H bond of the methylene group. Analysis of the spectra of all PE samples before and after irradiation showed that the width and position of the peaks remain unchanged, indicating the preservation of structural units. However, the ratio of peak intensities turned out to be sensitive to the direction of film stretching, which is expressed in an increase in the intensity of peaks in the region of C–C stretching and twisting of the CH_2_ group (spectra 2 and 5—along the stretching axis). When polymer chains become more oriented, in our case under uniaxial stretching, the dipole moments or polarizabilities of certain bonds become aligned with respect to the direction of radiation. This leads to an increase in the intensity of certain bands in Raman spectroscopy.

## 4. Discussion

In the course of the conducted studies of the influence of the type and modes of irradiation on PE and PTFE, common patterns of decreasing the rate of polymer return deformation after irradiation with high-energy electrons with an energy of 2 MeV were established.

The rate of polymer return deformation decreases after irradiation with electrons. According to the conducted studies, these changes are caused by the processes of creating free radicals under the influence of radiation in the studied polymers, which can break chemical bonds in the molecular chain, and also promote depolymerization.

Irradiation of polyethylene with krypton ions with an energy of 147 MeV leads to a significant decrease in the Vε, as well as irradiation of the material with high-energy electrons. Irradiation can cause cross-linking of polymer molecules, which makes them more rigid and less pliable. In the case of polyethylene, this can lead to the formation of a network structure, which increases rigidity, but reduces the ability to plastic deformation.

Optical micrographs taken with a DM 6000M (Leica) optical microscope show dark areas corresponding to surface contamination and volumetric heterogeneities in the sample. Elongated spindle-shaped areas—pores and microcracks—are observed on the samples after mechanical testing. The samples irradiated with high-energy electrons with an energy of 2 MeV after testing also demonstrate the appearance of microcracks and elongated pores. PTFE samples demonstrate a denser and less transparent structure compared to the PE film, which is evident from the photographs. In the case of non-irradiated PTFE before testing, heterogeneities are observed in the sample volume, probably associated with different densities. The photographs of the non-irradiated PTFE sample after mechanical testing show that the dark areas corresponding to denser areas have become more elongated along the film tensile axis. The sample irradiated with 2 MeV electrons in a vacuum before testing, similar to the unirradiated sample, demonstrates a non-uniform density distribution, which is expressed in dark areas in the micrographs. The irradiated PTFE sample after mechanical testing demonstrates more pronounced dark areas directed along the tensile axis, compared to the unirradiated sample after testing.

The analyzed PTFE films demonstrate a Raman signal typical for this material. Peaks in the region of 290 cm^−1^ and 384 cm^−1^ correspond to torsional and deformation vibrational modes of the CF_2_ group. The most intense peak at 733 cm^−1^ is caused by symmetric vibrations of compression and extension of the C–C bond, while a pronounced peak in the region of 1380 cm^−1^ corresponds to symmetric vibrations of compression and extension of the CF_2_ group. Weakly expressed peaks at 1217 and 1300 cm^−1^ are attributed to antisymmetric vibrations of the extension of CF_2_.

The analyzed PE films demonstrate characteristic Raman bands of polyethylene, as seen in Figure 8. Peaks in the region of 1064 and 1128 cm^−1^ correspond to compression and extension of the C–C bonds. The line at 1296 cm^−1^ is caused by the vibrational mode of twisting CH_2_ of the methylene group. The triplet in the region of 1440 cm^−1^ is caused by the bending of CH_2_ methylene groups. The broad peak in the region of 2723 cm^−1^ is a combination of several vibrational modes. Peaks in the range of 2800–3000 cm^−1^ are usually responsible for the compression and stretching of C–H bonds. In particular, the peaks at 2852 cm^−1^ and 2888 cm^−1^ correspond to symmetric and asymmetric stretching of methylene groups CH_2_, respectively. The shoulder-shaped peak in the region of 2930 cm^−1^ is also caused by the compression and stretching of the C–H bond of the methylene group. Analysis of the spectra of all PE samples before and after irradiation showed that the width and position of the peaks remains unchanged, which indicates the preservation of structural units.

Similar to the PE samples, the position and width of the peaks in polytetrafluoroethylene are preserved for all samples, indicating the preservation of structural units. The dependence of the intensity of some peaks on the orientation relative to the extension axis is also observed. It is important to note that, according to literature data, the intensity of the peak in the region of 1380 cm^−1^ often correlates with the deformations of the −CF_2_− group in the polymer chain. We have studied the return deformation of polymeric materials since this characteristic is often one of the main ones. For example, food film, with which we wrap food, should have this property, and studying this characteristic will make it possible to improve the properties of the films used. In the future, the authors plan to study the Vε of some polymeric materials under the influence of high temperatures (from 30 to 300 °C).

## 5. Conclusions

For a number of polymers, a study was conducted on the effect of the dose of electron and ion radiation, temperature and mechanical stress on the return and rate of return deformation. It was found that the structure of the polymers under study begins to change under the influence of electron and ion beams, as well as temperature and mechanical stress, which leads to a change in such mechanical properties as return deformation and rate of return deformation. The experimental data are explained within the framework of the proposed theoretical model. The area of practical use of the results obtained may be the development of packaging materials in the food industry.

## Figures and Tables

**Figure 1 materials-18-02316-f001:**
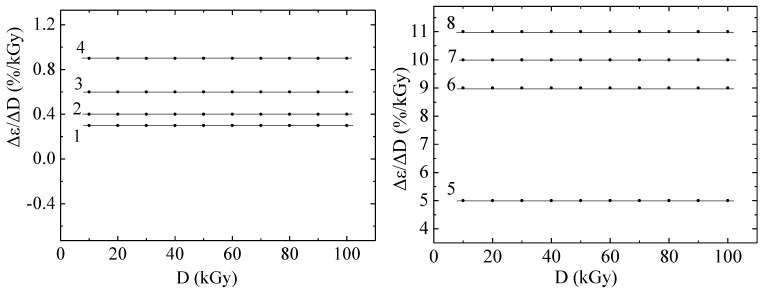
Dependence of the strain rate as a function of irradiation dose on the absorbed dose of irradiation with electrons with an energy of 2 MeV of polyethylene with a thickness of 50 μm at different temperatures and a static stress of 4.3 MPa. Dots—experiment; lines—calculations according to the model. 1—25; 2—30; 3—35; 4—40; 5—45; 6—50; 7—55; 8—60 °C.

**Figure 2 materials-18-02316-f002:**
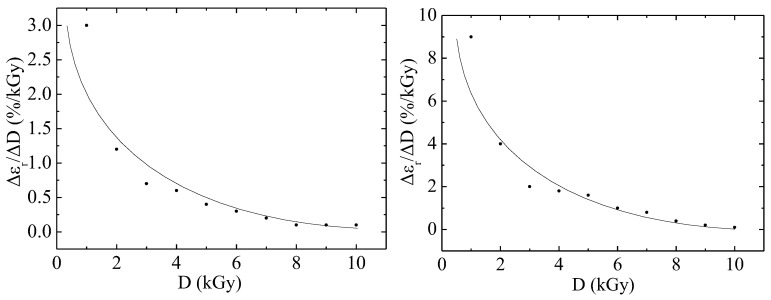
Dependence of the Vε as a function of irradiation dose on the absorbed dose of irradiation with electrons with an energy of 2 MeV for fluoroplastic films of 40 (**left**) and 100 (**right**) µm in size under a static stress of 23 MPa. Dots—experiment; lines—calculations according to the model.

**Figure 3 materials-18-02316-f003:**
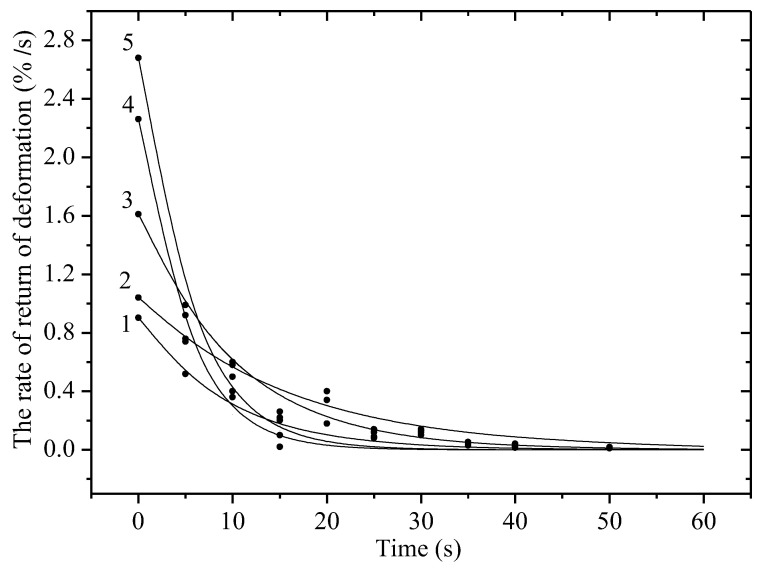
Dependence of the Vε on time of non-irradiated and irradiated with krypton ions with an energy of 147 MeV polyethylene film with a working length of 5 cm under a static stress of 80 MPa. Dots—experiment; lines—calculations according to the model. 1—non-irradiated sample; 2—1.5 × 10^6^; 3—1.6 × 10^7^; 4—5.0 × 10^7^; 5—10^9^ particles/s.

**Figure 4 materials-18-02316-f004:**
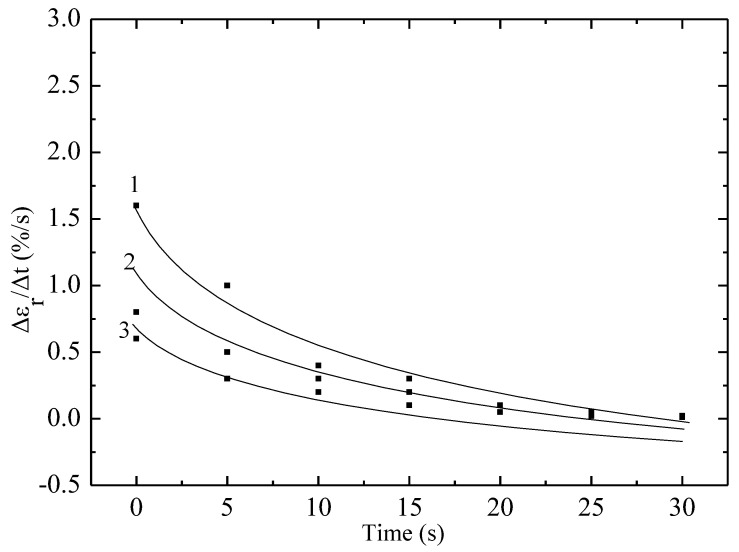
Dependence of the Vε on time of non-irradiated and electron-irradiated polytetrafluoroethylene film with a thickness of 40 μm and a working length of 5 cm under a static stress of 23 MPa. Dots—experiment; lines—calculations according to the model. 1—non-irradiated sample; 2—5; 3—10 kGy.

**Figure 5 materials-18-02316-f005:**
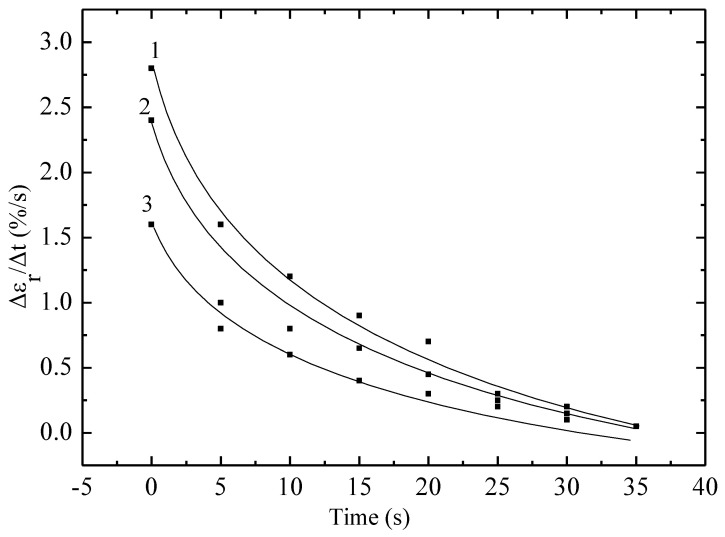
Dependence of the Vε on time of non-irradiated and irradiated with 2 MeV electrons polytetrafluoroethylene film with a thickness of 100 μm and a working length of 5 cm under a static stress of 23 MPa. Dots—experiment; lines—calculations according to the model. 1—non-irradiated sample; 2—5; 3—10 kGy.

**Figure 6 materials-18-02316-f006:**
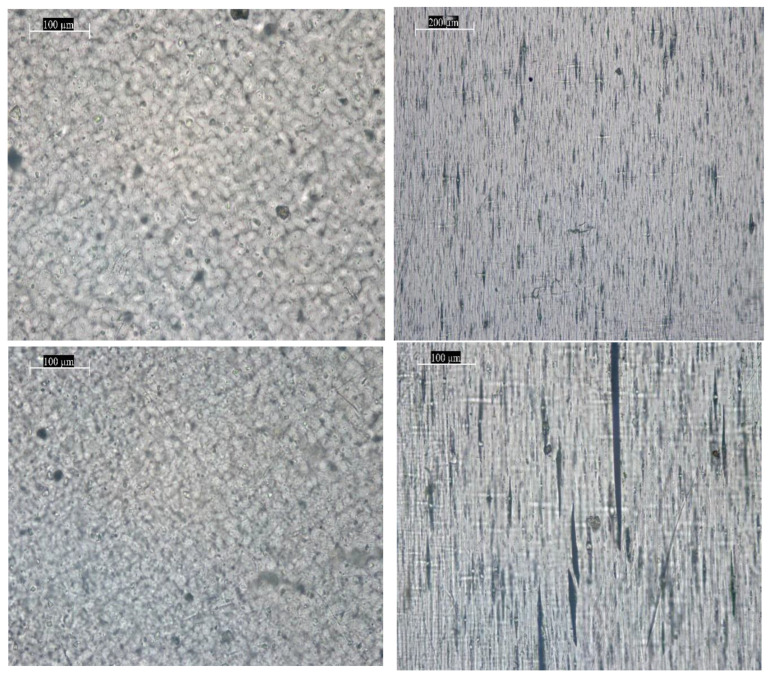
Photographs of different magnifications of the image of the surface morphology of the polyethylene film, obtained on an optical microscope DM 6000M (Leica). **Left**—before testing; **right**—after mechanical testing on a tensile testing machine; **top**—unirradiated; **bottom**—irradiated with 2 MeV electrons at a dose of 200 kGy. Magnification: 200 µm—×5000; 100 µm—×10,000.

**Figure 7 materials-18-02316-f007:**
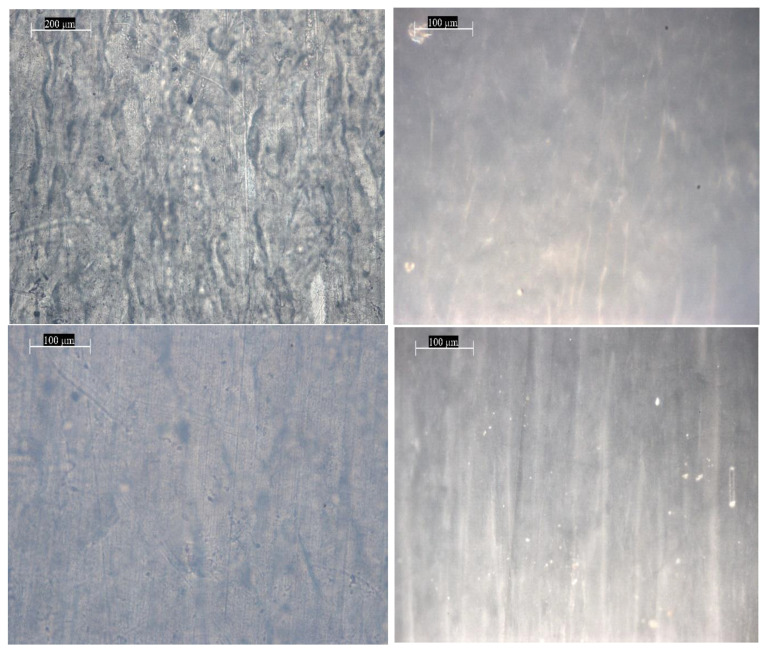
Photographs of different magnifications of the image of the surface morphology of polytetrafluoroethylene film, obtained on a DM 6000M optical microscope (Leica). **Left**—before testing; **right**—after mechanical testing on a tensile testing machine; **top**—unirradiated; **bottom**—irradiated with 2 MeV electrons at a dose of 10 kGy. Magnification: 200 µm—×5000; 100 µm—×10,000.

**Figure 8 materials-18-02316-f008:**
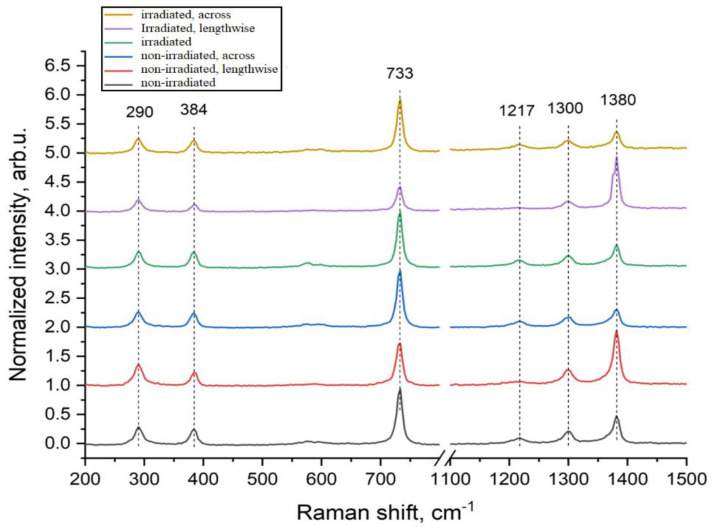
Raman spectra of unirradiated and irradiated polytetrafluoroethylene samples with 2 MeV electrons at a dose of 10 kGy before and after testing.

**Figure 9 materials-18-02316-f009:**
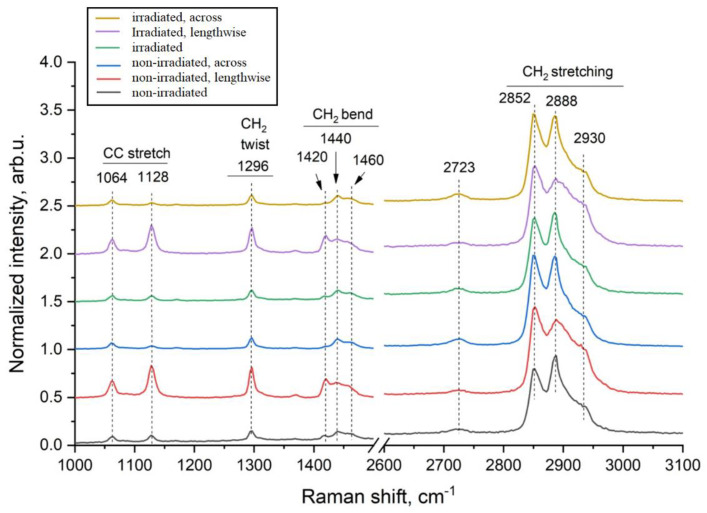
Raman spectra of unirradiated and irradiated polyethylene samples with 2 MeV electrons at a dose of 200 kGy before and after testing.

**Table 1 materials-18-02316-t001:** The characteristics of the materials for this study, the type and modes of irradiation.

Material	Thick, μm	Type and Modes of Irradiation
PE	40	High-energy electrons with an energy of 2 MeV and a current intensity of 0.16 μA/cm^2^ at doses of 10, 30, 50, 70 and 100 kGy
	100	High-energy electrons with an energy of 2 MeV and a current intensity of 0.16 μA/cm^2^ at a dose of 200 kGy
	23	Krypton ions with an energy of 147 MeV at doses of 1.5 × 10^6^; 3–1.6 × 10^7^; 4–5.0 × 10^7^; 5–10^9^ particles/s
PTFE	100	High-energy electrons with an energy of 2 MeV and a current intensity of 0.16 μA/cm^2^ at doses of 1, 2, 3, 4, 5, 6, 7, 8, 9 and 10 kGy

## Data Availability

The original contributions presented in this study are included in the article. Further inquiries can be directed to the corresponding author.

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
