# Peer review of "Investigation into Time-Dependent Deformation and Recovery Rates in Non-Irradiated and Irradiated Polymers"

_materials, 2025, doi:10.3390/ma18102316_

Round 1

Reviewer 1 Report

Comments and Suggestions for Authors

This study aims to investigate the effect of the radiation dose on the rate of deformation and the rate of return deformation 66 of polymers under constant load. Here is the comments:

Title: It is difficult to read as there are lots of "of", please consider to make a shorter and condensed title.

Abstract: The structure is too loose which reduce the readability for this paper. You can simply say one or two sentence of the background of your study. Why it is so important for you to investigate that, then mention the aims of your research and the application of your results. 

Introduction: It is too brief and short for only 3 paragraph. You need to add more details on what have been done previously from other researchers and background study. There are only 11 references in the introduction part, which is way too little. It should be a comprehensive background search before your study. 

Experimental method and material: It is better to add the table to conclude what material and methods has been use. In the paragraph, you can explain more details on why you choose this method and material. For the material part, it is better to mention the supplier of the material and testing instrument. If you use the standard tests, you also need to state the standard number for those tests. For the experimental method, please also add some images to show how to test your samples. 

Results: Figure statement should be place underneath of the images. There are few notes has been added between the figure statement and images. You should add those notes in the paragraph or in the image if needed. Now, the layout is unusual and difficult to read.

Discussion and Conculsion: It should be separated into two sections but not one. For the discussion part, you need to compare your results with previous literature. Is your results same as the other researchers or different from them? What does it like that? It gives comprehensive analytical results to the reader. 

Comments on the Quality of English Language

Title: It is difficult to read as there are lots of "of", please consider to make a shorter and condensed title.

Author Response

Response 1:  We thank the reviewer for the helpful suggestion. The title of the article was changed to the following.

Comments 2: Abstract: The structure is too loose which reduce the readability for this paper. You can simply say one or two sentence of the background of your study. Why it is so important for you to investigate that, then mention the aims of your research and the application of your results.

Response 2:  The abstract has been completely revised and written from a new perspective based on your recommendations.

Comments 3: Introduction: It is too brief and short for only 3 paragraph. You need to add more details on what have been done previously from other researchers and background study. There are only 11 references in the introduction part, which is way too little. It should be a comprehensive background search before your study.

Response 3: The authors have made an additional review of similar works and the list of references has increased accordingly.

Comments 4: Experimental method and material: It is better to add the table to conclude what material and methods has been use. In the paragraph, you can explain more details on why you choose this method and material. For the material part, it is better to mention the supplier of the material and testing instrument. If you use the standard tests, you also need to state the standard number for those tests. For the experimental method, please also add some images to show how to test your samples.

Response 4: We have described the used method of testing samples for tension ASTM-D882. In addition, all the parameters of the RU-50 tensile testing machine settings are described.

Comments 5: Results: Figure statement should be place underneath of the images. There are few notes has been added between the figure statement and images. You should add those notes in the paragraph or in the image if needed. Now, the layout is unusual and difficult to read.

Response 5: We understand your comment. We have corrected everything so that it is convenient to read.

Comments 6: Discussion and Conculsion: It should be separated into two sections but not one. For the discussion part, you need to compare your results with previous literature. Is your results same as the other researchers or different from them? What does it like that? It gives comprehensive analytical results to the reader. 

Response 6:  The point is that these polymeric materials have not been studied in terms of return deformation and return deformation rate, so we cannot compare them with the works of other scientists, as we believe that this is incorrect. We have irradiated them with ions and electrons and are comparing the change in characteristics with each other.

Reviewer 2 Report

Comments and Suggestions for Authors

This manuscript studies the deformation and return deformation behavior of PE and PTFE films subjected to electron and ion irradiation. The authors have experimentally investigated the strain rate and its recovery with varying radiation doses, film thicknesses, and irradiation sources. The authors applied exponential models based on a cascade-probability method (CPM) to describe the results. Optical microscopy and Raman spectroscopy were employed to support the mechanical findings.

Below are my specific comments

  1. The authors should provide a more thorough introduction and explanation of the CPM model, including its physical basis, advantages for this study, and any prior work that has successfully employed this model in relative fields.
  2. The CPM fitting parameters and accuracy (R2) should be included. These values should be either included in the figure or summarized in a separate table.
  3. The original tensile stress–strain curves should be presented in the main figures or supplementary information. Furthermore, since the manuscript discusses irradiation-induced cross-linking and rigidity, a summary of mechanical properties before and after irradiation would be informational.
  4. In the optical microscopy analysis, the authors should quantify the microcrack or pore length in irradiated and non-irradiated samples.
  5. Polymer films like PE and PTFE typically exhibit anisotropic mechanical properties in the machine direction and cross direction. The authors should clarify the orientation of testing. It will also be beneficial to include additional study on how orientation affects the mechanical response before and after irradiation.

Author Response

Comments 1: The authors should provide a more thorough introduction and explanation of the CPM model, including its physical basis, advantages for this study, and any prior work that has successfully employed this model in relative fields

Response 1: We have corrected the introduction and provided information about the cascade-probability method, and also referred to our work where this method was previously used.

Comments 2: The CPM fitting parameters and accuracy (R2) should be included. These values should be either included in the figure or summarized in a separate table.

Response 2: We have added the parameters that are present in the formulas of the models proposed by the authors below the description of the formula. There are not so many of them that we need to indicate the data in the table.

Comments 3: The original tensile stress–strain curves should be presented in the main figures or supplementary information. Furthermore, since the manuscript discusses irradiation-induced cross-linking and rigidity, a summary of mechanical properties before and after irradiation would be informational.

Response 3: A summary of the mechanical properties before and after irradiation was written by the authors after reading your comment.

Comments 4: In the optical microscopy analysis, the authors should quantify the microcrack or pore length in irradiated and non-irradiated samples.

Response 4: In the optical microscopy analysis, the authors should quantify the microcrack or pore length in irradiated and non-irradiated samples.

Comments 5: Polymer films like PE and PTFE typically exhibit anisotropic mechanical properties in the machine direction and cross direction. The authors should clarify the orientation of testing. It will also be beneficial to include additional study on how orientation affects the mechanical response before and after irradiation.

Response 5: The direction of stretching of the samples has been clarified and written in the article after reading your comment.

5. Additional clarifications

The responses of the authors of the article in each document and the corrections in the article are indicated in red font.

Reviewer 3 Report

Comments and Suggestions for Authors

The manuscript addresses the effect of ion and electron irradiation on deformation and return strain rates of polyethylene (PE) and polytetrafluoroethylene (PTFE) under static load. The experimental setup is well-described, and the combination of mechanical testing, surface microscopy, and Raman spectroscopy provides a multi-angle perspective on polymer modification under irradiation.

The paper is overall well-structured and contributes to the field of polymer irradiation, especially by analyzing return deformation rate, which is often overlooked in favor of traditional strength parameters. The proposed cascade-probability model is appropriate and adequately fits the experimental data.

However, some points require clarification or stylistic improvement to increase clarity, cohesion, and reader engagement.

  1. The abstract is somewhat technical. Suggest simplifying to emphasize what was done, why it's important, and what was found
  2. lines 21–67: The statement “there are no works on the study of the effect of the radiation dose on the rate of deformation and the rate of return deformation” is too strong. Suggest softening to:
    "While many studies examine mechanical properties of irradiated polymers, detailed analysis of deformation and recovery kinetics under static load remains limited."
  3. Figure 1-3 - Add more detail to captions. Specify what each curve represents without needing to refer back to the text (e.g., define "flow of dose", describe sample geometry).
    Clarify how the parameters ε₀ and D₀ (or t₀) were obtained – were they fitted or based on previous work?
  4. The cascade-probability method is referenced, but only briefly explained. A short paragraph justifying why this model suits polymer strain recovery would be helpful for readers outside the radiation damage field.
  5. Section - mechanical results - The distinction between "rate of deformation" and "rate of return deformation" is central but not always clearly maintained. → Consider adding a visual or schematic summarizing the loading-unloading cycle and what exactly is measured in each phase.
  6. On Raman analysis - The interpretation is informative, but the implications could be expanded. For instance, what does preservation of structural bands say about macromolecular stability under irradiation? Could the increase in peak intensity be linked to molecular orientation or crystallinity?

  7. Figures 6–8: Improve caption clarity. Currently, the captions lack context (e.g., define which spectra correspond to before/after testing).
  8. My sugestion - Add a table summarizing all tested samples (material, thickness, dose, irradiation type, static load) to help readers keep track.
  9. Use consistent units and labels. 
  10. The term "rate of return deformation" is repeated very often. Consider alternating with terms like “recovery rate”, “strain recovery”, or “post-deformation response” where appropriate.
  11. Minor Grammar Fixes:
    • “with the flow of the dose” - should be “as a function of irradiation dose”
    • “polymeric materials irradiated by electrons” - “polymeric materials irradiated with electrons”
    • "The parameter became almost equal to zero" - "approached zero"

Author Response

Comments 1: The abstract is somewhat technical. Suggest simplifying to emphasize what was done, why it's important, and what was found

Response 1: The abstract has been completely revised and written from a new perspective based on your recommendations.

Comments 2: lines 21–67: The statement “there are no works on the study of the effect of the radiation dose on the rate of deformation and the rate of return deformation” is too strong. Suggest softening to:
"While many studies examine mechanical properties of irradiated polymers, detailed analysis of deformation and recovery kinetics under static load remains limited."

Response 2: Your comment has been taken into account and we have corrected this issue.

Comments 3: Figure 1-3 - Add more detail to captions. Specify what each curve represents without needing to refer back to the text (e.g., define "flow of dose", describe sample geometry).
Clarify how the parameters ε₀ and D₀ (or t₀) were obtained – were they fitted or based on previous work?

Response 3: Details to the signatures have been added to the text, and the methods for obtaining these parameters have been clarified. All these points are written in the corrected version of the article.

Comments 4: The cascade-probability method is referenced, but only briefly explained. A short paragraph justifying why this model suits polymer strain recovery would be helpful for readers outside the radiation damage field.

Response 4: The cascade-probability method (CPM), invented by us, is used in radiation physics to calculate the concentration of radiation defects, including in materials. It is based on the cascade-probability function (CPF), which has the meaning of the probability of a primary particle to reach a depth h with energy E0 after the number of collisions i. In this paper, the simplest CPF is used. Within the framework of this method, we have explained many experimental data:

1. When polymers are irradiated, chains are mainly broken and simple defects are formed that affect mechanical properties, including deformation.

2. When materials are irradiated with electrons with an energy of 2 - 6 MeV, electrical resistance, internal friction and positron annihilation at low and normal temperatures.

Comments 5: Section - mechanical results - The distinction between "rate of deformation" and "rate of return deformation" is central but not always clearly maintained. → Consider adding a visual or schematic summarizing the loading-unloading cycle and what exactly is measured in each phase.

Response 5: The experiments measured the rate of deformation and return deformation of the samples. The rate of deformation was calculated using the graphs of the dependence of deformation on time under static stress. And the rate of return deformation was calculated after the load on the samples was removed and they began to take their previous shape. The corresponding explanations are given in the text.

Comments 6: On Raman analysis - The interpretation is informative, but the implications could be expanded. For instance, what does preservation of structural bands say about macromolecular stability under irradiation? Could the increase in peak intensity be linked to molecular orientation or crystallinity?

Response 6: Thank you for your valuable comments. The corresponding explanations have been added to the text.

Comments 7: Figures 6–8: Improve caption clarity. Currently, the captions lack context (e.g., define which spectra correspond to before/after testing).

Response 7: In these figures we have given all the necessary characteristics, and some of them are shown in the captions to the figures.

Comments 8: My sugestion - Add a table summarizing all tested samples (material, thickness, dose, irradiation type, static load) to help readers keep track.

Response 8: Thank you for the suggestion! We tried to adjust the text so as to do without the table.

Comments 9: Use consistent units and labels. 

Response 9: We have taken your comment into account and corrected the units of measurement.

Comments 10: The term "rate of return deformation" is repeated very often. Consider alternating with terms like “recovery rate”, “strain recovery”, or “post-deformation response” where appropriate.

Response 10: For the term rate of return deformation, the notation Vε was introduced in the Introduction section. This notation is used further in the text. In this way, we fulfilled the requirement of uniformity of the term.

Comments 11: Minor Grammar Fixes:

·        “with the flow of the dose” - should be “as a function of irradiation dose”

·        “polymeric materials irradiated by electrons” - “polymeric materials irradiated with electrons”

·        "The parameter became almost equal to zero" - "approached zero"

Response 11: Thank you for your comment! The text has been amended in accordance with your recommendations.

Round 2

Reviewer 1 Report

Comments and Suggestions for Authors

Experimental : "The following materials were the object of research in this work. 74
1. 40 μm thick PE irradiated with high energy electrons with an energy of 2 MeV 75 and a current intensity of 0.16 μA/cm2 at doses of 10, 30, 50, 70 and 100 kGy 76
2. 100 μm thick PE irradiated with high energy electrons with an energy of 2 MeV 77 and a current intensity of 0.16 μA/cm2 at a dose of 200 kGy 78
3. 40 and 100 μm thick PTFE irradiated with high energy electrons with an energy of 79 2 MeV and a current intensity of 0.16 μA/cm2 at doses of 1, 2, 3, 4, 5, 6, 7, 8, 9 and 10 kGy 80
4. 23 μm thick PE irradiated with krypton ions with an energy of 147 MeV at doses 81 of 1.5 ·106; 3 1.6 ·107; 4 5.0·107; 5 109 particles/s." This information should be presented in a table or in a paragraph but not in a point form. Also, it should not be at the beginning of ethe experimental design. 
4. Discussion and Conclusions: You need to separate this into two sections. One section is a discussion and the last section is conclusion. OR you can put the discussion in the results and the subtitle should be results and discussion.
Lines 163-164: "Irradiation of the 163 material with ions leads to a significant decrease in the Vε, as does irradiation of the 164 material with high energy electrons." Please double-check the grammar of this sentence. 
Lines 118-119: These few lines of content is that belong to the next paragraph or in the figure? It is very confusing to have those lines underneath a graphic. If it belongs to the figure, you can put it on the right side or left side of the figure. The same thing happens to the figures 1-4.

Comments on the Quality of English Language

Please double-check the grammar of the manuscript.

Author Response

Dear Editor,

On behalf of all the co-authors, I would like to express our gratitude to you and the reviewer for your insightful comments on our original submission. We believe that the revised version addresses all the observations made. The changes are highlighted in red font, and additional details are provided below. Furthermore, we have directly addressed the comments in the revised manuscript and made all necessary corrections accordingly.

In accordance with the journal’s guidelines for submitting a revised manuscript, we have uploaded the following documents to the system:

  1. Detailed Response to Reviewers
  2. Cover Letter
  3. Declaration of Interest Statement
  4. Manuscript
  5. Revised Manuscript (Unmarked)
  6. Revised Manuscript with Marked Changes

Many thanks for your attention

Best Regards,

Marat. N. Niyazov

R = reviewer Comments

A = authors’ reply

Reviewer

R1 Experimental : "The following materials were the object of research in this work. 74
1. 40 μm thick PE irradiated with high energy electrons with an energy of 2 MeV 75 and a current intensity of 0.16 μA/cm2 at doses of 10, 30, 50, 70 and 100 kGy 76
2. 100 μm thick PE irradiated with high energy electrons with an energy of 2 MeV 77 and a current intensity of 0.16 μA/cm2 at a dose of 200 kGy 78
3. 40 and 100 μm thick PTFE irradiated with high energy electrons with an energy of 79 2 MeV and a current intensity of 0.16 μA/cm2 at doses of 1, 2, 3, 4, 5, 6, 7, 8, 9 and 10 kGy 80
4. 23 μm thick PE irradiated with krypton ions with an energy of 147 MeV at doses 81 of 1.5 ·106; 3 1.6 ·107; 4 5.0·107; 5 109 particles/s." This information should be presented in a table or in a paragraph but not in a point form. Also, it should not be at the beginning of ethe experimental design. 

A1 We offer our deepest apologies for not having made these corrections earlier. Now the required Table with the necessary information about the samples and irradiation has been inserted into the test.

R2 Discussion and Conclusions: You need to separate this into two sections. One section is a discussion and the last section is conclusion. OR you can put the discussion in the results and the subtitle should be results and discussion.

A2 Thank you for your valuable comment. The corresponding amendments have been made to the text.

R3 Lines 163-164: "Irradiation of the 163 material with ions leads to a significant decrease in the Vε, as does irradiation of the 164 material with high energy electrons." Please double-check the grammar of this sentence. 

A3 The sentence has been corrected to match the content.

R4 Lines 118-119: These few lines of content is that belong to the next paragraph or in the figure? It is very confusing to have those lines underneath a graphic. If it belongs to the figure, you can put it on the right side or left side of the figure. The same thing happens to the figures 1-4.

A4 Thank you for your comment. The captions from the general text have been inserted into the captions to the figures.

Reviewer 2 Report

Comments and Suggestions for Authors

The authors have thoroughly addressed all of my comments

Author Response

Thank you very much for taking the time to review this manuscript.